# Disentangling the link between maternal influences on birth weight and disease risk in 36,211 genotyped mother–child pairs
Jaakko T. Leinonen [1], FinnGen*, Matti Pirinen [1,2,3] & Taru Tukiainen [1] ✉

Epidemiological studies have robustly linked lower birth weight to later-life disease risks. These observations may reflect the adverse impact of intrauterine growth restriction on a child's health. However, causal evidence supporting such a mechanism in humans is largely lacking. Using Mendelian Randomization and 36,211 genotyped mother-child pairs from the FinnGen study, we assessed the relationship between intrauterine growth and five common health outcomes (coronary heart disease (CHD), hypertension, statin use, type 2 diabetes and cancer). We proxied intrauterine growth with polygenic scores for maternal effects on birth weight and took into account the transmission of genetic variants between a mother and a child in the analyses. We find limited evidence for contribution of normal variation in maternally influenced intrauterine growth on later-life disease. Instead, we find support for genetic pleiotropy in the fetal genome linking birth weight to CHD and hypertension. Our study illustrates the opportunities that data from genotyped parent-child pairs from a population-based biobank provides for addressing causality of maternal influences.

The relationship between birth weight and disease risk has been extensively studied in epidemiological settings, revealing associations with adverse health consequences both for lower and higher birth weights[1–8]. Of particular note is the connection between low birth weight and cardiometabolic diseases[9,10]. This connection has been speculated to arise from long-lasting changes in metabolic programming due to intrauterine growth restriction, referring to poor fetal growth[9,10]. This concept has led to the hypothesis of developmental origins of health and disease (DOHaD) suggesting that several non-communicable diseases originate in early development—during prenatal life, in an unfavorable intrauterine environment, or in early childhood[11]. The causal evidence in support of the DOHaD mechanism in humans has been, however, largely lacking, and conflicting[3,12,13]. Consequently, although the relationship between birth weight and disease risk is clear at the epidemiological level, the underlying causes behind these associations have been a subject for debate for decades[3].

Mendelian randomization (MR) has become a popular method for assessing causal relationships between an exposure and an outcome[14]. In short, MR utilizes genetic variants robustly associated with the exposure to test whether the same variants have consistent effects on the outcome. MR strategies using genetic variants of the mother associated with birth weight of the child therefore offer a means to assess the potential causal influences of intrauterine growth as approximated by the birth weight of the child[12,14,15].

Variation in birth weight has a large genetic component (SNP-based heritability ($h^2_{SNP}$) being ~40%, of which a fifth can be specifically allocated to maternal genetic variation), making it an amenable trait for genetic studies[15]. However, since many different factors, including both the maternal and fetal genome, influence the size of the baby, using MR in assessing the DOHaD hypothesis has some important prerequisites[3,15–18]. First, using genetic factors specifically reflecting maternal influences on birth weight as instrumental variables in MR is crucial[12,19], as these can plausibly reflect intrauterine growth as opposed to the variants with strict fetal effects. Secondly, a standard two-sample MR using the maternal genotype only can lead to biased causality estimates as it fails to account for the 50% correlation between the maternal and fetal genotypes, therefore violating the exclusivity assumption of MR[12,20]. To overcome this issue, data sets with genotyped mother-child pairs are of great value, as they allow for blocking the path through the child's genome by adjusting the analysis with information on the child's genetic variants at the loci tested.

Recent studies utilizing these MR principles and mother-child pairs suggest that small genetic effects on intrauterine growth (IUG) have only limited effects on child's cardiometabolic risk factors[12,15]. These studies,

[1]Institute for Molecular Medicine Finland (FIMM), HiLIFE, University of Helsinki, Helsinki, Finland. [2]Department of Public Health, Faculty of Medicine, University of Helsinki, Helsinki, Finland. [3]Department of Mathematics and Statistics, University of Helsinki, Helsinki, Finland. *A list of authors and their affiliations appears at the end of the paper. ✉e-mail: taru.tukiainen@helsinki.fi

based on mother-child pairs from the Norwegian HUNT study ($N = 26{,}057$)[12] and the UK Biobank ($N = 3886$)[15], have reported lack of strong effects of IUG on cardiometabolic risk factors such as lipid and glucose levels and hypertension. To our knowledge, comparable studies are limited, primarily due to the scarcity of suitable genotyped parent–child cohorts[21]. Additionally, it remains uncertain whether these results might extend to the development of cardiovascular disease.

Population-based biobank data may offer new possibilities to conduct MR studies of maternal exposures[12,22], as these data typically capture also familial relationships due to their large scale of sampling. For instance, within the 430,897 participants of the FinnGen study (release 10)[23], corresponding to 8% of the Finnish population, there are 67,986 parent-offspring and 63,428 full sibling pairs with genotype and extensive longitudinal health registry information available. A particular advantage of the parent–child pairs contained within FinnGen, including 36,211 mother-child and 31,775 father-child duos, is the comparatively advanced age of the children (mean 46.3 years, SD 14.2), allowing for examination of disease endpoints that manifest later in life.

Here, we apply an established MR framework[12] to the 36,211 mother-child pairs from the FinnGen study to examine how polygenic scores (PGS) for maternal influences on birth weight associate with five common health outcomes in the children ($N$ cases = 996–6150, Fig. 1), while considering the effects from both the maternal and child genomes in the same model. Extending the previous findings on biomarkers[12,15], we find no evidence for the role of IUG, as proxied by genetic scores for maternal effects on birth weight, in determining child's disease risks. Rather, we show that the scores of the same variants in the fetal genome associate with CHD and hypertension, an effect that is also detected in the 31,775 father-child pairs, and in analyses based on the 63,428 sibling pairs from FinnGen. Collectively, these findings suggest that genetic pleiotropy in the child is largely accountable for the epidemiological links between birth weight and disease.

## Results
### Polygenic scores predict birth weight in FinnGen
We first constructed polygenic scores (PGSs) for birth weight using the findings of the latest genome-wide association study (GWAS) of individual's own ($N = 321{,}223$) and offspring birth weight ($N = 230{,}069$ mothers) that applied structural equation modeling to dissect the birth-weight-associated genetic markers ($N = 209$) into those with maternal only, fetal only, or shared effects[15]. Using these GWAS results, we built six PGSs that capture different degrees of maternal and fetal influences on child's birth weight. To exclusively model the maternal contribution to birth weight, we used two PGSs reflecting strictly maternal effects on birth weight: a score based on 29 lead SNPs with only maternal effects on birth weight (M-SPECIFIC) and a respective genome-wide score (M-GW). We supplemented these with two other lead SNP-based scores (M-ALL and MF-ALL) that contain variants with both maternal and fetal effects on birth weight, and with two scores reflecting specifically fetal effects on birth weight (F-SPECIFIC and F-GW; "Methods" section and Supplementary Fig. 1a).

As we aimed to use these PGSs as instrumental variables within a MR framework to assess causality, our first step was to confirm that the PGSs predict the measured birth weight in FinnGen. To this end we utilized a subset of FinnGen participants with available birth weight measurements, based on the national birth registry (FinnGen release 10 $N = 39{,}578$; mother-child pairs $N = 9257$; father-child pairs $N = 5740$).

All the birth weight PGSs showed statistically significant effects on birth weight when calculated from an individual's own genotypes in the full FinnGen data ($P < 0.05$; Supplementary Data 1). In mother–child pairs, M-ALL, MF-ALL, F-SPECIFIC, and F-GW PGSs were associated with both child and own birth weight, as expected since these scores contain variants with both maternal and fetal effects on birth weight. Importantly, however, the two PGSs based on variants with maternal effects, M-SPECIFIC and M-GW, showed strictly mother-specific effects on child's birth weight. In other words, only the mother's score influenced the child's birth weight (at the significance level $P < 0.05$) when including both the mother's and child's M-SPECIFIC or M-GW PGSs in a multiple regression model (Supplementary Data 1).

To further validate the predictive values of the PGSs, we computed them for fathers and children in the father-child pairs from FinnGen. Here, as expected given these scores should capture maternal effects, neither M-SPECIFIC nor M-GW of the father associated with child's birth weight (Supplementary Data 1). However, for M-GW, we noted a significant negative association ($P < 0.05$) of the father's PGSs on the child's birth weight after including the child's PGS in the model[24,25]. This can indicate

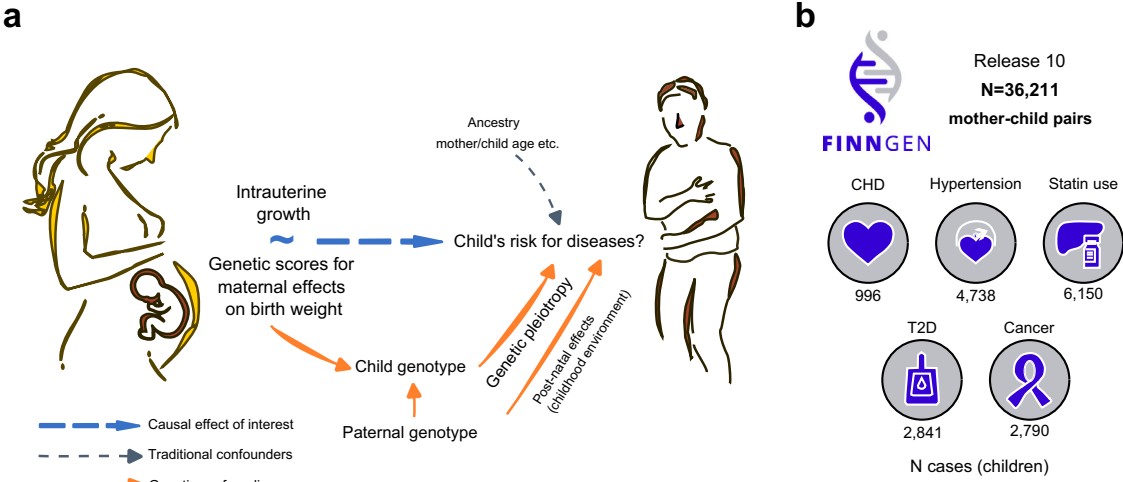

**Fig. 1 | Illustration of the Mendelian Randomisation framework applied in mother-child pairs with a summary of the FinnGen dataset. a** Illustration of the Mendelian Randomisation (MR) framework to assess the relationship between intrauterine growth (IUG) and child's disease risk, when IUG is approximated by the child's birthweight. This MR analysis tests whether a mother's genetic score for a child's birth weight is associated with the child's disease risk later in life. MR relies on the assumption that mother's genetic score does not affect the child's disease risk through any pathway other than IUG. Since a child inherits 50% of genetic variants from the mother, approximately half of the variants contributing to mother's genetic score are transmitted to the child. Theoretically, the variants that affect the child's size in the mother can have different functions when transmitted to the child's genome (genetic pleiotropy). Therefore, it is vital to account for this genetic sharing by conditioning the MR analyses on the child's genetic score. **b** Summary information on the FinnGen data set, with the case numbers in the children from the mother-child pairs indicated for each disease. CHD coronary heart disease, T2D type 2 diabetes.

both the presence of a potential collider bias and the fact that a genome-wide score may contain an excess of variants that are not specific to the trait studied (genetic pleiotropy). We therefore chose to focus on M-SPECIFIC as our main instrument for maternally influenced IUG for further analyses.

## The birth weight PGSs associate with disease risk in the whole FinnGen dataset

The connections between birth weight and several diseases are well established at both the epidemiological and genetic levels, also in Finnish datasets[1,4–7,12,15,26,27]. For the FinnGen participants, measured birth weight data are available only for participants born after 1987. These participants are too young to manifest, e.g., cardiac symptoms, making it unfeasible to study epidemiological patterns in this subset. However, echoing the epidemiological observations, we detected many associations between the birth weight PGSs computed from a person's own genotypes and the disease outcomes in the whole FinnGen dataset ($N = 412,176$; Supplementary Data 2). While these population-level analyses indicate clear relationships between lower birth weight and higher risk, e.g., for cardiovascular disease, they do not allow the assessment of DOHaD as the underlying causal mechanism, as the influence of the birth weight variants are examined in one's own genome.

## Mother's PGS for maternal effects on child birth weight shows no association to child's disease risk after taking into account the child's PGS

To address the potential causality of DOHaD, we therefore focused on understanding how the PGSs specific for maternal effects on birth weight, as a proxy for intrauterine growth, associate with disease risk in the 36,211 mother-child pairs in FinnGen. A key component of the MR framework applied here is to include both the mother's and the child's PGS in the analyses to account for the 50% correlation between these PGSs (Fig. 1 and Supplementary Fig. 1d)[12]. Failure to adjust the analyses with the child's own PGS can lead to a spurious association between maternal PGS and child's diseases, as shown by our simulations (Supplementary Data 3 and Supplementary Fig. 2). Instead, in case of true intrauterine effects, any association between a mother's PGS and child's disease risk remains unchanged when adding information from the child's PGS to the model (Supplementary Data 3).

When analyzing the effects of the mother's and child's PGS separately, (i.e., including only one of the PGSs into the model) both the maternal and child's own PGS showed several statistically significant associations with diseases ($P < 0.05$). For example, higher mother's M-SPECIFIC PGS was associated with a reduced risk for CHD (OR = 0.92, [95% CI 0.86–0.97], $P = 0.0089$) and statin use, (OR = 0.96 [95% CI 0.93–0.99], $P = 0.016$), and increased risk for cancer (1.04 [95% CI 1.00–1.08], $P = 0.049$). Child's own PGS was similarly associated with these disease outcomes (for M-SPECIFIC OR = 0.87 [95% CI 0.81–0.92], $P = 1.5e-05$ (CHD); OR = 0.96, [95% CI 0.93–0.99] $P = 0.024$ (Hypertension); OR = 0.97, [95% CI 0.94–0.99], $P = 0.022$ (Statin use) and OR = 1.05, [95% CI 1.01–1.09], $P = 0.015$ (Cancer; Fig. 2 and Supplementary Data 4).

However, when taking into account both the mother's and the child's PGS simultaneously, the maternal PGS, which explained child's birth weight irrespective of child's PGS, no longer displayed associations with child's disease risks ($P > 0.05$; Fig. 2 and Supplementary Data 4). Here, matching the expectations from simulations, in two instances, the effect sizes of the maternal M-SPECIFIC PGS were reduced compared to the unadjusted models (from 0.97 to 0.99 for hypertension, and 0.92 to 0.98 for CHD; Fig. 2 and Supplementary Data 4). Instead, in the same analyses child's own M-SPECIFIC PGS remained statistically significantly associated with both diseases, with comparable effect sizes as in the unadjusted analyses OR = 0.87 [95% CI 0.81–0.94], $P = 0.00047$ for CHD and OR = 0.96 [95% CI 0.92–0.99], $P = 0.032$ for hypertension).

While the results for CHD and hypertension aligned with the expectations of exclusive fetal effects (Fig. 2 and Supplementary Fig. 2), for the three other traits the patterns of associations were more complex, though

non-significant in models including both maternal and child PGSs (Fig. 2 and Supplementary Data 4). In case of statin use and cancer we observed significant associations ($P < 0.05$) in the models including only one PGS. Yet, these became non-significant in the combined model, which may reflect reduced statistical power or joint maternal and fetal effects on these outcomes. For T2D, the maternal and fetal point estimates from the joint model were to the opposite directions, though with large standard errors (Fig. 2 and Supplementary Data 4).

## Results from the other birth weight PGSs support exclusive fetal effects on disease risk

After the main analyses, we ran additional tests to probe the robustness of our findings. Although our focus was on the effect of an unweighed lead-SNP-based PGS specifically tagging maternal effects on birth weight (M-SPECIFIC), results from the other PGSs with a maternal component (M-GW, M-ALL, and MF-ALL) supported the concept that birth weight–disease associations are largely driven by the effects of genetic variants in the child's genome (Supplementary Fig. 3a–d and Supplementary Data 4). None of the mother's PGSs were associated with disease risk in the children after considering the child's own PGS ($P > 0.05$), and the maternal effect sizes were usually reduced, as expected based on the simulations where the child's own genome conferred the risk (Supplementary Figs. 2 and 3, and Supplementary Data 3 and 4). Also, the fetal PGSs (F-SPECIFIC and F-GW) showed association to disease only through the child's own genome. The effects of maternal and fetal birth weight PGSs in the children were generally similar in direction, with higher birth weight protecting from CHD, hypertension, and statin use. However, for T2D, the effect directions were the opposite between the maternal and fetal PGSs, with a higher fetal PGS protecting from, and a higher PGSs for maternal effects increasing T2D risk (Supplementary Fig. 3 and Supplementary Data 4).

## Sensitivity analyses in the mother-child pairs support the lack of maternal effects on child disease risk

As the FinnGen mother-child data contains mothers ($N = 5483$) that have more than one child included in the dataset, we tested how excluding such non-independent pairs affects the detection of the maternal effects on disease. We noted that the lack of significant maternal effects remained true regardless of whether we limited our dataset to include only one child per mother ($N = 28,582$, Supplementary Data 5). Similarly, using a stringently filtered dataset where only a maximum of 4th degree relatives from both mothers and children were included ($N = 22,454$), or adjusting the results with a PGS for gestational duration did not support presence of maternal effects (Supplementary Data 5). The only exception to the rule was statin use for which we detected nominal support for maternal effects in the stringently unrelated mother-child pairs (Supplementary Data 5). Importantly, we detected that the effect size for a mother's PGSs did not change in a statistically significant manner in any of these sensitivity analyses (Supplementary Fig. 4a).

## Data from the father-child pairs supports predominantly fetal effects on disease risks

We next followed up on these findings in the 31,775 father–child pairs (child mean age 47.6 years [SD 13.8]) available in FinnGen. Here, similarly as in the birth weight prediction (see Supplementary Data 1), the association of a father's PGS with child's disease risks is unexpected, as no intrauterine mechanisms are in play, and any association would thus point to effects from the postnatal environment or confounding by assortative mating. Yet, we would expect any effects of the fetal genome to be present also in the father-child pairs. In line with these expectations, we detected little evidence for paternal effects for the PGSs tagging maternal effects on birth weight. The exception was paternal M-GW associating with statin use (OR 1.06 [95% CI 1.02–1.10], $P = 0.0021$), raising the possibility of a postnatal contribution to this phenotype or reflecting the more complex nature of a genome-wide PGSs in terms of tagging genetic pleiotropy (Supplementary Fig. 5b and Supplementary Data 6). Instead, we repeatedly observed an

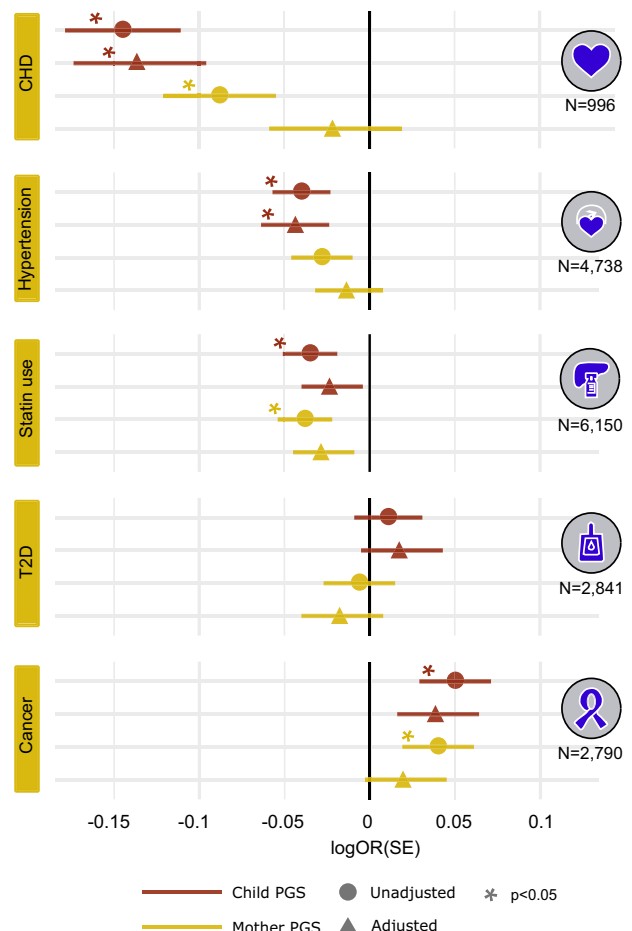

**Fig. 2 | Association of a PGS for maternal effects on birth weight with child's disease risk in FinnGen mother-child pairs.** The figure illustrates the association between the child's own PGS (red) and mother's PGS (yellow) with the child's disease risk in the mother-child pairs. Round dots represent the effect estimates (in logOR units) from analyses using only one PGS (either maternal or fetal M-SPE-CIFIC), while triangles indicate analyses including M-SPECIFIC PGSs from both mothers and their children. The lines around the effect estimates mark standard error (SE). An asterisk (*) denotes a statistically significant association ($P < 0.05$) between a PGS and disease risk in the children. The numbers beneath the disease symbols indicate case numbers per disease. Coronary heart disease (CHD) $N = 996$; Hypertension $N = 4738$; Statin use $N = 6150$; type 2 diabetes (T2D) $N = 2841$; Cancer $N = 2790$.

association between the child's own genetic scores for maternal effects on birth weight (M-SPECIFIC) and disease risk also in the father-child pairs, with comparable effect estimates as in the mother-child pairs, e.g., OR = 0.91 [95% CI 0.84–0.99], $P = 0.021$ for CHD and OR = 0.95 [95% CI 0.91–0.99], $P = 0.012$ for hypertension (Supplementary Fig. 5a and Supplementary Data 6).

### Analysis of FinnGen full sibling pairs strengthens the conclusions from parent–child data

We finally sought validation of these findings using the 63,428 full sibling pairs (mean age 63.9 years [SD 16.4]) from FinnGen. Leveraging sibling pairs discordant for the outcomes of interest enables another nuanced way to determine whether a PGS for maternal effects on birth weight affects disease risk through the maternal or fetal genome. As siblings share their maternal genetic effects on birth weight, any association of the maternal PGS with disease risk seen in these sibling analyses should solely reflect the child's own genetic effects. To test this, we assessed associations of the birth weight PGSs in a matched case-control setting using same-sex sibling pairs,

discordant for disease status (case numbers ranging from 6016 CHD cases to 11,505 statin users in the final datasets). Applying a conditional logistic regression model, we found that one's own PGSs for maternal effects on birth weight was associated with a reduced risk of CHD and hypertension in the full sibling pairs (M-SPECIFIC OR = 0.93 [95%CI 0.89–0.98], $P = 0.0049$ and OR = 0.94 [95% CI 0.89–0.99], $P = 0.017$, for CHD and hypertension, respectively; Supplementary Data 7 and Supplementary Fig. 6). Similarly, fetal-specific birth weight variants displayed an effect on CHD through one's own genome (OR = 0.95 [95% CI 0.89–0.99], $P = 0.011$ for F-SPECIFIC). In contrast, besides the fetal genome-wide PGS (F-GW) association to statins (OR = 0.92 [95% CI 0.89–0.96], $P = 5.0\text{e-}05$), none of the PGSs containing variants with maternal effects on birth weight were associated with statin use or cancer risk in siblings. However, we observed distinct connections for the birth weight PGSs to T2D, with the direction of the effect again depending on whether the PGS tags maternal or fetal effects (OR = 1.06 [95% CI 1.01–1.11], $P = 0.023$ for M-GW and OR = 0.89 [95% CI 0.85–0.93], $P = 1.6\text{e-}06$ for F-GW; Supplementary Data 7 and Supplementary Fig. 6). Overall, these sibling comparisons further support the interpretation that the effects of birth weight PGSs on disease are primarily mediated by genetic pleiotropy of the birth weight variants in the child's genome.

### Power calculations

We conducted simulations to assess the magnitude of maternal and fetal effects we were sufficiently powered to detect in the 36,211 mother–child pairs for each endpoint (see Methods; Supplementary Data 3). These analyses indicated that when including both maternal and child PGSs in the analysis we had adequate power (≥80%) to detect true maternal effects (OR per 1 SD change in PGS) ranging from >1.11 for CHD to >1.05 for statin use (Supplementary Data 3 and Supplementary Fig. 7). These analyses estimate the power given the case frequencies in the available mother-child dataset. We, however, note that although many of the children included in the study are old enough to manifest, e.g., cardiac symptoms, a quarter of the children from the mother–child pairs were less than 34.4 years old at the end of follow-up, which is a limiting factor in the dataset. In addition, our power calculations do not consider how much variance a PGS explains of birth weight, and hence direct translation of our estimates to clinical effects is more challenging.

### Discussion

The relationship between birth weight and disease risks has been intensively studied for several decades, and many different theories have been proposed to explain their connections. In this study, we set out to examine the causal mechanisms between lower birth weight and later-life health using large-scale genetic data. In particular, we aimed to understand the proposed role of intrauterine growth, i.e., the mechanism of the DOHaD hypothesis, as a determinant of the child's disease risks. To this end, we used a specifically designed Mendelian randomization (MR) framework in a large sample of Finnish genotyped mother-child pairs. This framework overcomes many limitations of regular MR related to assessing maternal influences, allowing for a more accurate estimation of potential causal effects of intrauterine growth on a child's risk for diseases[12]. Here intrauterine conditions are proxied using genetic markers with strict maternal effects on a child's birth weight, and the independent contribution of the maternal genome is assessed by blocking the direct transmission of alleles by adjusting the analysis using the child's genetic score at the same loci.

Taking advantage of a unique data set of 36,211 genotyped mother-child pairs available within the FinnGen study, we applied the MR framework on five health registry endpoints epidemiologically associated with birth weight. Overall, we did not find support for a strong connection between maternally influenced intrauterine growth and offspring later life disease. Rather, it seems that the same genetic variants that, when present in mothers affect a child's birth weight, once inherited by the child, have independed effects on the child's disease risks. Our findings thus point to the links of lower birth weight and diseases occuring largely due to genetic

pleiotropy in the child's genome. This main conclusion was supported by the numerous sensitivity analyses, including using different birth weight PGSs, data from father-child pairs, and data from outcome discordant sibling pairs.

Our study builds upon previous research investigating the DOHaD mechanisms using mother-child and biomarker data from biobanks, including the UK Biobank ($N = 3886$)[15] and the HUNT cohort ($N = 26,057$)[12]. In this study, we expand upon these analyses by using a larger number of mother-child pairs and examining the associations between maternal effects on birth weight and five binary disease outcomes sourced from nationwide health registries. Our decision to focus on binary outcomes, while potentially reducing statistical power, uniquely positions us to directly explore the link between genetically determined maternal effects on birth weight and disease manifestations, moving beyond the previous investigations of disease risk factors.

Given the lack of statistically significant maternal contribution to child's disease risks, our results suggest that modest changes in intrauterine growth may have limited effects on diseases such as CHD and hypertension compared to the effects of the child's own genome. Echoing the findings from Moen et al.[12], our data shows that the same SNPs that associate with a child's birth weight in mothers exert independent genetic effects on disease risks when transmitted to children. Reflecting this genetic pleiotropy, the disease risks associated with the birth weight scores were consistently more closely linked to the effects from the fetal rather than the maternal genome. Clear examples of such cases were CHD and hypertension risks, for which there was consistently very little evidence of any intrauterine effects in play after taking into account the child PGSs. This finding was further strengthened by sibling analyses, where we found that the sibling with a higher PGS for maternal effects on birth weight had a reduced risk for CHD and hypertension, despite the maternal genetic effect for birth weight being shared between the siblings. Our findings thus support the idea that blood pressure and birth weight are connected through the alleles that first reduce the child's birth weight when present in the mother, and then increase the child's blood pressure when present in the child, as previously suggested in smaller samples[15]. Based on our findings, a similar mechanism appears to hold for CHD risk.

Although our data allows us to conclude that the effects on disease risks mediated by the birth weight PGSs act principally through the child genome, we detected a couple of instances where the results were less clear-cut. In several instances, we associated the birth weight PGSs with one's own or child's risk of receiving statin medication yet could not always confidently exclude potential maternal or paternal contributions to these associations. This was especially true in case of the genome-wide PGSs for birth weight (M-GW and F-GW). It thus remains possible that some genetic factors affecting birth weight may be related to statin use through a postnatal, e.g., behavioral, component. Alternatively, the mixed results might partly reflect the complex pleiotropy tagged especially by the genome-wide PGSs. However, the result should be followed up and validated in additional datasets.

Finally, our data highlight associations between birth weight and the risk of T2D, with previous studies backing both intrauterine and genetic mechanisms behind this connection[13,15,28]. Results from those PGSs that were based on alleles with predominantly fetal effects on birth weight (MF-ALL, F-SPECIFIC, and F-GW) strongly support the fetal insulin hypothesis, stating that the same genetic factors that increase birth weight in the fetal genome also protect against T2D[6,7]. However, in contrast, a higher genome-wide PGS for maternal effects in birth weight (M-GW) was consistently associated with an increased T2D risk, both in mother–child pairs and in sibling analyses. It thus seems that the genetic variants influencing birth weight can have rather complex effects on lifetime T2D risk, depending on their means of action.

Despite the many benefits of using biobank data to explore the connections between maternal traits and child's disease outcomes, our study and datasets have limitations. In this study, we used genetic variants associated with birth weight as quantitative traits to proxy intrauterine growth in a population-based sample. The birth weight PGSs that were used as instrumental variables in our analyses showed clear effects on birth weight and disease risk at population level. Hence, we posit that these are valid instruments to explore the known epidemiological connections between birth weight and disease risks later in life under the MR setting. We nonetheless stress that all the PGSs were based on common genetic variants and explain only a proportion of the total variance in birth weight in FinnGen (1 SD change in our genetic instrument (M-SPECIFIC) corresponded to ~41 g change in birth weight). We thus acknowledge that in this study we may not explicitly model, for example, severe intrauterine growth restriction resulting from external factors.

Further, instead of reflecting solely intrauterine growth, the PGSs may be partly related to normal variation in child's size, for example, due to gestational duration, though our results, when adjusting for the PGS for gestational length, suggest that controlling for this has negligible effect on the disease associations[17,29]. In addition, though we used established MR principles, and therefore could test for evidence of the causality of maternal effects on birth weight on disease risk, we have not performed formal MR to provide accurate effect estimates for the effects of birth weight. Also, we assume a simple monotonic relationship between the child's disease risk and the PGS for birth weight, which for some phenotypes can be suboptimal, as both low and high birth weight can increase the risk of same disease[3]. Finally, our power analyses also indicate that we have likely been limited to detecting maternal effects that are relatively large. We also note that the inclusion of children that are below the expected age of onset for some of the included endpoints such as CHD may affect the power to detect effets in our main analyses.

The key medical implication from this work is that modest changes in intrauterine growth during pregnancy are unlikely to have large effects on child's disease risk in later life. In contrast to the role of intrauterine conditions, our findings support a model wherein the genetic factors within the maternal genome that influence the child's birth weight, contribute to the child's disease risks only when transferred to the child's genome. Overall, our study demonstrates how MR in genotyped mother-child pairs is a sound and powerful method to evaluate how maternal and fetal exposures relate to child's health. We envision that large-scale population-based biobanks, such as FinnGen applied here, can enhance the power for such studies and that they will allow for testing many other hypotheses in the field.

## Methods
### Ethics statement
All patients and control participants in FinnGen provided informed consent for biobank research, based on the Finnish Biobank Act. Research cohorts collected prior to the start of FinnGen (in August 2017) were collected based on study-specific consents and later transferred to the Finnish biobanks after approval by Valvira, the National Supervisory Authority for Welfare and Health. Recruitment protocols followed the biobank protocols approved by Valvira. The Coordinating Ethics Committee of the Hospital District of Helsinki and Uusimaa (HUS) approved the FinnGen study protocol Nr HUS/990/2017.

The FinnGen study is approved by Finnish Institute for Health and Welfare (THL), approval number THL/2031/6.02.00/2017, amendments THL/1101/5.05.00/2017, THL/341/6.02.00/2018, THL/2222/6.02.00/2018, THL/283/6.02.00/2019, THL/1721/5.05.00/2019, Digital and population data service agency VRK43431/2017-3, VRK/6909/2018-3, VRK/4415/2019-3 the Social Insurance Institution (KELA) KELA 58/522/2017, KELA 131/522/2018, KELA 70/522/2019, KELA 98/522/2019, and Statistics Finland TK-53-1041-17. The Biobank Access Decisions for FinnGen samples and data utilized in FinnGen Data Freeze 10 include: THL Biobank BB2017_55, BB2017_111, BB2018_19, BB_2018_34, BB_2018_67, BB2018_71, BB2019_7, BB2019_8, BB2019_26, BB2020_1, BB2021_65, Finnish Red Cross Blood Service Biobank 7.12.2017, Helsinki Biobank HUS/359/2017, HUS/248/2020, HUS/150/2022 §12, §13, §14, §15, §16, §17, §18, and §23, Auria Biobank AB17-5154 and amendment #1 (August 17 2020) and amendments BB_2021-0140, BB_2021-0156 (August 26 2021,

Feb 2 2022), BB_2021-0169, BB_2021-0179, BB_2021-0161, AB20-5926 and amendment #1 (April 23 2020)and it's modification (Sep 22 2021), Biobank Borealis of Northern Finland_2017_1013, 2021_5010, 2021_5018, 2021_5015, 2021_5023, 2021_5017, 2022_6001, Biobank of Eastern Finland 1186/2018 and amendment 22§/2020, 53§/2021, 13§/2022, 14§/2022, 15§/2022, Finnish Clinical Biobank Tampere MH0004 and amendments (21.02.2020 & 06.10.2020), §8/2021, §9/2022, §10/2022, §12/2022, §20/2022, §21/2022, §22/2022, §23/2022, Central Finland Biobank 1-2017, and Terveystalo Biobank STB 2018001 and amendment 25th Aug 2020, Finnish Hematological Registry and Clinical Biobank decision 18th June 2021, Arctic biobank P0844: ARC_2021_1001.

## FinnGen study

The FinnGen study (https://www.finngen.fi/en) is an on-going research project that utilizes samples from a nationwide network of Finnish biobanks and digital health care data from national health registers[23]. The goal of the project is to produce genomic data with linkage to health register data for over 500,000 biobank participants nationwide. The majority of the samples have been gathered from six university hospital biobanks. In the present study, we included samples from 430,897 biobank participants with genotypes available (FinnGen release 10). The samples are linked to national hospital discharge (available from 1968), death (1969–), cancer (1953–) and medication reimbursement (1964–) registries. Additional registries include national birth registry (1987-) containing, e.g., data for birth weight, and the registry on medication purchases (1995-). Currently, after sample pruning and quality control the release 10 of the dataset contains phenotypes for 412,176 participants (181,869 men and 230,307 women, median age of 62.9 years), representing roughly 8% of the Finnish population. Due to the sample ascertainment and selection procedures, the cohort has clear advantages over some other population-based sample collections. For example, given that most samples are from hospital biobanks, FinnGen includes an excess of disease cases. However, due to the same reasons FinnGen should not be considered as an epidemiologically representative dataset[23].

The parent-offspring relationships (total $N = 72,465$) used in this study had been inferred from the genetic data by the FinnGen analysis team with KING software using the suggested tresholds for calling first-degree relatives[30]. After sample pruning, e.g., excluding duplicate samples and ethnic outliers, and removing suggested parent–child relationships with age difference between samples <15 years, we were left with 67,986 parent–child relationships in the dataset (36,211 mother-child pairs, 31,775 father-child pairs with phenotype data available for analysis). We identified altogether 28,582 unique mothers. The majority of the parents were born before 1970, with an average of 1.26 children per parent (Supplementary Fig. 8). The mean age of the mothers was 70.0 years, the mean age of the fathers 71.2 years, and the mean age of the children 46.3 years, at the end of the followup period. The mean age of the children from the mother-child pairs was 45.0 years (SD 14.5). Similarly to the parent-offspring relationships, siblings from the FinnGen data ($N = 66,668$) were identified through the KING analysis. After QC we were left with 63,428 full sibling pairs with an average age of 63.8 years (SD 16.8).

## Disease endpoints and phenotype data

Birth weight data was available for 39,578 participants in FinnGen R10, born after 1987. For our main analyses, we included the following five predefined endpoints from the FinnGen registry team: coronary heart disease (I9_CHD), hypertension (FG_HYPERTENSION), statin use (RX_STATIN, a proxy for high cholesterol levels), type 2 diabetes (E4_DM2_STRICT) and cancer (C3_CANCER_EXALLC). The disease case number in FinnGen R10 ranged from 46,959 CHD cases to 144,672 statin users (~11.3% to 35.1% of the dataset). In the mother-child pairs, the $N$ case range for children was from 996 CHD cases to 6150 statin users (corresponding to 2.8% and 16.7% of the dataset), echoing the observation that many, but not all children are old enough to manifest, e.g., cardiac symptoms. In the sibling analyses, for each studied endpoint we

included a subset of the identified full sibling pairs, matched by sex and discordant for the disease in question ($N$ cases range 6016 CHD to 11,505 statin users). The exact case numbers for all analyses are available in Supplementary Data 4-7. More detailed phenotype descriptions and definitions and summary data for these phenotypes for whole FinnGen R10 dataset are available from risteys.finngen.fi.

## Construction of the maternal and fetal polygenic scores for birth weight

We constructed altogether four different polygenic scores (PGSs) to study the relationships between child's birth weight and later life disease risks in the FinnGen cohort. All PGSs were based on GWAS data from the Early Growth Genetics (EGG) Consortium[15]. The EGG Consortium data included summary statistics for GWAS of own ($N = 321,223$) and child birth weight ($N = 230,069$), partitioning the genetic effects into maternal and fetal components.

Two genetic scores were designed to capture specifically maternal effects on birth weight, to proxy intrauterine growth. We first constructed a similar unweighted lead SNP-based PGS as used in Moen et al.[12], by summing up the number of genome-wide significant birth weight increasing alleles per individual. The unweighed score for maternal effects on birth weight (M-SPECIFIC) was built based on 32 SNPs identified in GWASs on own and offspring birth weight. Upon partitioning genetic effects into maternal and fetal components using structural equation modeling, these 32 SNPs were reported to have specifically a maternal effect on birth weight[15]. Secondly, we calculated a genome-wide polygenic score (PGSs) for maternal effects on birth weight (M-GW), based on the summary statistics of a GWAS on offspring birth weight, adjusting for fetal effects using an extension of structural equation modeling, and a respective genome-wide PGS for fetal effects on birth weight (F-GW)[15].

In addition, we constructed two additional scores partially reflecting maternal effects, of which M-ALL was based on 72 SNPs from the birth weight GWASs, consisting of the 32 SNPs with specifically maternal effects on birth weight, 27 SNPS with directionally concordant maternal and fetal effects on birth weight, and 15 SNPs with directionally opposing maternal and fetal effects[15]. Finally, we calculated an unweighed score (MF-ALL) based on the beforementioned 72 SNPs, 64 SNPs with fetal-only effects, and 71 unclassified SNPs[15]. We finally supplemented these scores with a lead SNP-based PGS tagging specifically fetal effects (F-SPECIFIC), based on 68 SNPs from the birth weight GWAS classified as having specifically fetal effects.

The unweighed scores were calculated with plink2 (www.cog-genomics.org/plink/2.0/)[31]. For the unweighed maternal scores, in FinnGen, we found data for 29, 68 and 201 SNPs respectively, whereas the fetal score was based on 62 SNPs (Supplementary Data 8-11). The relationships of the studied PGSs are illustrated in Supplementary Fig. 1a, b. The use of unweighed PGSs has been previously argued to be a more valid measurement for maternal effects on birth weight since the exact effect sizes for the SNPs on the intrauterine growth are unknown[12]. We chose to also include a genome-wide score based on maternal allelic weights in our study since this had more power to explain variance in birth weight compared to the lead SNP-based scores (Supplementary Data 1). For comparison, we constructed a similar genome-wide score based on fetal effects on birth weight. The genome-wide scores were calculated with PRS-CS[32] using the FinnGen PRS pipeline (https://github.com/FINNGEN/CS-PRS-pipeline), filtering data from GWAS of maternal effects on birth weight adjusted for child's effects to include only HapMap3 SNPs. The downside of using the genome-wide scores, that take the maternal allele weights for both the mother and the child, is that we cannot as accurately control for the pleiotropic effects of the variants in the child's genome as when using lead SNP-based scores. In theory, relying only on the genome-wide PGSs thus might lead to an excess of false positives (PGS of mother associates with child's disease through child's genome rather than through maternal effects during the pregnancy).

We standardized all PGSs into z-scores and reported their effects per SD-unit in our analyses. The EGG Consortium samples based on which the

PGSs were built are largely independent of the target samples in the Finn-Gen, although we note that ~4.7% of the EGG Consortium GWAS participants are of Finnish ancestry[15]. We could identify some FinnGen samples that have been included in calculating the GWAS summary statistics for EGG, but these overlapping samples make up only ~0.4% of the FinnGen mother-child cohort (from NFBC66 and NFBC86, N = 146). We consider their potential effects on the results negligible. The PGSs used in this study are derived from a trans-ancestral GWAS meta-analysis, primarily composed of individuals of European ancestry, with the variants identified in the GWAS explaining a substantial proportion of variance in birth weight in the Norwegian MoBa cohort[15]. In line of these findings from another Northern European cohort, we observed that the constructed unweighed and genome-wide PGSs showed transferability also to the FinnGen cohort, capturing the desired effects on birth weight (Supplementary Data 1).

## Statistics and reproducibility

In our primary analyses, we tested for associations between mothers' PGSs for maternal effects on birth weight (M-SPECIFIC) and child's disease, adjusting for the same PGS computed for the child. The principles of the MR framework are illustrated in Fig. 1. The analyses were run using logistic regression (glm function with family ="binomial" in R), and were adjusted for child age, first 10 principal components of genetic structure and genotyping batch. The results from these analyses were obtained as logarithm of odds ratio (logOR) and its standard error (SE) per standard deviation (SD) increase in the PGS. For our main tables we transformed the logOR values and the corresponding SEs or 95% confidence intervals (95%CI) to odds ratio (OR) scale for more intuitive interpretation. All analyses were performed using R Statistical Software (v4.3; R Core Team 2023). Given the overlap between most of the disease endpoints and the direct relationships between the PGSs, we did not adjust for multiple testing and we use $P < 0.05$ as the significance threshold. We complemented our main analyses by including associations from five other PGSs (M-GW, M-ALL, MF-ALL, F-SPECIFIC and F-GW) into the study. As further sensitivity analyses, we a) performed similar analyses using only one (oldest) child per mother (N = 28,582), b) kept only the oldest mother and her oldest child from an extended family (N = 22,454 mother-child pairs with 3th degree relatives and closer for both mothers and children removed from the analysis) based on relationships identified in the KING analysis[30], or c) adjusted for polygenic scores for maternal effects on gestational length[17]. The polygenic scores for gestational length were constructed based on GWAS results from the EGG Consortium, using summary statistics from maternal GWAS meta-analysis of gestational duration based on 151,987 women[17]. The genome-wide PGS for gestational length used in the analyses was calculated similarly as the genome-wide scores for birth weight, using PRS-CS and the FinnGen PRS pipeline.

Besides the analyses in the mother-child pairs, we utilized the identified father-child pairs (N = 31,775) to test for the presence of potential postnatal effects and to further control for potential familial effects. Finally, we followed up our findings in FinnGen sibling pairs (N = 63,428), using conditional logistic regression (clogit function from the "survival" package (https://CRAN.R-project.org/package=survival) in a matched case-control setup, selecting full sibling duos of same sex but discordant for disease status into the analyses. Such a setup allows for natural control of maternal genetic effects on birth weight, which are shared between the siblings, permitting estimation of the fetal effects of the PGSs.

We acknowledge that some associations between birth weight and disease show a J-shaped curve (both low and very high birth weight increase disease risk compared to more typical birth weight)[3]. The associations to higher birth weight are anyhow visible only with very high birth weights that are likely outside the variation that our genetic instruments capture, and we therefore chose to include only monotonic effects in our analyses.

## Power calculations

We conducted simulations to estimate the statistical power of our framework to capture maternal effects on child's endpoints under different combinations of true effects of mother's and child's PGSs (Supplementary Software 1)[33]. We ran 1000 simulations for each combination of maternal and child effects on a given endpoint. In each simulation, we:

1. used the mvtnorm R package (http://mvtnorm.R-forge.R-project.org) to generate N = 36,211 samples from two-dimensional normal distribution (means 0, variances 1, correlation 0.5) to reflect the birth weight PGSs for the FinnGen mother-child pairs,
2. using these distributions, the case number for the studied endpoint, and chosen maternal and child effects constructed an exponential risk function, an inverse of logistic regression, where the prevalence of the disease matched the observed prevalence in the FinnGen data, and the effect sizes are on a logOR scale.

$$Risk = \frac{\exp\left(\alpha + \beta\, child * PGS\, child + \beta\, mother * PGS\, mother\right)}{1 + \exp\left(\alpha + \beta\, child * PGS\, child + \beta\, mother * PGS\, mother\right)}$$

3. randomly sampled a case-control status for each child according to the child's risk value, and
4. then regressed the sampled case-control vector jointly on the mother's and child's PGSs using logistic regression.

The estimate for power was obtained as the proportion of simulations where the p-value for the coefficient of interest (mother's or child's PGS) was below the p-value threshold of 0.05. Power calculations were conducted in R (version 4.2.3 (2023-03-15)).

## Reporting summary

Further information on research design is available in the Nature Portfolio Reporting Summary linked to this article.

## Data availability

Full genetic and clinical data from FinnGen is available for researchers by application (https://www.finngen.fi/en/access_results). Details of FinnGen core endpoints can be found at risteys.finngen.fi. The source data for results figures is available in Supplementary Data: for Fig. 2—Supplementary Data 4; Supplementary Fig. 2—Supplementary Data 3; Supplementary Fig. 3—Supplementary Data 4; Supplementary Fig. 4—Supplementary Data 4–6; Supplementary Fig. 5—Supplementary Data 6; and Supplementary Fig. 6—Supplementary Data 7.

## Code availability

The code for power analyses is available as Supplementary Software 1[33]. The full genotyping and imputation protocol for FinnGen is described at https://doi.org/10.17504/protocols.io.nmndc5e.

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

## Acknowledgements

We thank all FinnGen participants, principal investigators, laboratory personnel and data management teams. The FinnGen project is funded by two grants from Business Finland (HUS 4685/31/2016 and UH 4386/31/2016) and the following industry partners: AbbVie Inc., AstraZeneca UK Ltd, Biogen MA Inc., Bristol Myers Squibb (and Celgene Corporation & Celgene International II Sàrl), Genentech Inc., Merck Sharp & Dohme LCC, Pfizer Inc., GlaxoSmithKline Intellectual Property Development Ltd., Sanofi US Services Inc., Maze Therapeutics Inc., Janssen Biotech Inc, Novartis AG, and Boehringer Ingelheim International GmbH. Following biobanks are acknowledged for delivering biobank samples to FinnGen: Auria Biobank (www.auria.fi/biopankki), THL Biobank (www.thl.fi/biobank), Helsinki Biobank (www.helsinginbiopankki.fi), Biobank Borealis of Northern Finland (https://www.ppshp.fi/Tutkimus-ja-opetus/Biopankki/Pages/Biobank-Borealis-briefly-in-English.aspx), Finnish Clinical Biobank Tampere (www.tays.fi/en-US/Research_and_development/Finnish_Clinical_Biobank_Tampere), Biobank of Eastern Finland (www.ita-suomenbiopankki.fi/en), Central Finland Biobank (www.ksshp.fi/fi-FI/Potilaalle/Biopankki), Finnish Red Cross Blood Service Biobank (www.veripalvelu.fi/verenluovutus/biopankkitoiminta), Terveystalo Biobank (www.terveystalo.com/fi/Yritystietoa/Terveystalo-Biopankki/Biopankki/) and Arctic Biobank (https://www.oulu.fi/en/university/faculties-and-units/faculty-medicine/northern-finland-birth-cohorts-and-arctic-biobank). All Finnish Biobanks are members of BBMRI.fi infrastructure (www.bbmri.fi). Finnish Biobank Cooperative -FINBB (https://finbb.fi/) is the coordinator of BBMRI-ERIC operations in Finland. The Finnish biobank data can be accessed through the Fingenious® services (https://site.fingenious.fi/en/) managed by FINBB. T.T. was funded by Sigrid Juselius Foundation (https://sigridjuselius.fi/en/), the HiLIFE Fellows Program, and the Research Council of Finland (grant number 315589). M.P. was supported by the Research Council of Finland (https://www.aka.fi/en/) grant 338507, Research Council of Finland Center of Excellence in Complex Disease Genetics grant 352795, and the Sigrid Juselius Foundation. Open access funded by Helsinki University Library.

## Author contributions

J.T.L. and T.T. conceived the study. J.T.L. and T.T. performed data analyses. T.T. formulated the power analyses. M.P. provided statistical support. FinnGen provided data. J.T.L. and T.T. drafted the original manuscript. J.T.L., M.P., and T.T. critically revised the draft manuscript. All authors approved the final version of the manuscript.

## Competing interests

The authors declare no competing interests

## Additional information

## FinnGen

**Taru Tukiainen** ⑮[1] ✉

A full list of members and their affiliations appears in the Supplementary Information.

