## [Peer Review File · Communications Biology]

Reviewers' comments:

Reviewer #1 (Remarks to the Author):

Leinonen and colleagues have conducted a Mendelian randomization study to assess the potential causal effects of intrauterine growth on offspring's health outcomes later in life using individual level data from mother-child pairs from FinnGen (n = 36,211). By assessing the effects of genetic scores of birth weight (with either maternal or maternal and fetal effects) using maternal and offspring genotypes, the authors provide supporting evidence that the fetal genome, rather than the intrauterine environment, has long-term effects on health. By using this specific framework, the authors suggest the effect of the fetal polygenic scores of birth weight on adult-life health outcomes reflect genetic pleiotropy rather than causality.

This is a carefully-conducted study that provides valuable information on the lack of causal effects of intrauterine growth on offspring's health outcomes in an expanded set of mother-child pairs compared to previous studies. The evidence provided is solid, although it would benefit from including instruments composed exclusively of genetic variants with fetal only effects on birth weight and assessing its effects on adult health outcomes.

I have some minor comments:

1. I understand the framing of the study with regards to DOHaD, but I wonder about defining the genetic scores as instruments for "intrauterine growth restriction", instead of "intrauterine growth". The publication where these scores were first defined suggest these variants reflect normal variations in birth weight, rather than pathologic cases, and is reflected in the inclusion of individuals with an own birth weight within normal ranges, for instance in the UK Biobank.
2. I also found the naming of the genetic scores slightly confusing, and had to go back and forth between methods/results to check what the acronym referred to. Naming these by the number of variants included, instead of using a descriptive name might be part of the confusion.
3. I appreciate the authors validating the genetic instruments used and assessing the statistical power, which is important. Can the authors comment on how the predictive capacity of the different genetic scores can impact such power analyses?
4. While the main conclusion refers largely to the lack of association of the maternal birth-weight increasing alleles on offspring's health outcomes, additional evidence suggesting that fetal genetic scores affect such outcomes would strengthen the conclusion. This could be done by using a genetic score of variants only associated with birth weight (using the weighted effects) in the full set of FinnGen (either individual level data or 2-sample MR), or using the set of variants with only fetal-effects (unweighted effects) in mother-child pairs (combined analysis). This will boost power in analyses where the fetal PGS was not significant after adjusting for the maternal one (lines 150-151).
5. In addition to the genetic association, I find it would be informative to describe the observational associations in a similar set of individuals. I acknowledge this is not necessary to answer the main study question.
6. In line with the reflection in line 319-320, would this be something to consider in the analysis/discussion of power? From Supp Figure 7, one can see that $\sim 1/4$ of the sample are ~ 30 years old.
7. The paragraph starting in line 232 refers mostly to data provided as supplementary material. Would

it be worth commenting on these results in the main results' section?

8. I find line 262 (conclusion) somewhat confusing: "Our findings, aligned with earlier..". Later in this sentence, the word "instead" (referring to the previous assertion) and independent effects (referring to maternal transmitted variants) are confusing to me.

9. The construction of the polygenic score of gestational duration is not explained in methods

10. The genetic scores have a different naming in Supp Table 1/2 than in the main text.

11. I appreciate very much that the authors added the code for running the simulations. I have rerun it, and noticed that there is a missing closing bracket in the 3rd row by the end of the code. Otherwise, it works perfectly well and power analyses are easy to reproduce.

Reviewer #2 (Remarks to the Author):

This study found limited evidence of a causal relationship between intrauterine growth restriction (IUGR) and non-communicable diseases such as coronary heart disease (CHD) and hypertension in the FinnGen biobank. Instead, the findings suggest that genetic pleiotropy in the child is mostly responsible for the association of birth weight and the five investigated diseases. Using 4 different polygenic scores (PGS) and sensitivity analyses, the conclusions and interpretation are solid and highly convincing. The abstract, introduction and discussion are well written, and previous literature clearly exposes the current knowledge about the developmental origins of health and disease (DOHaD hypothesis) conundrum. These results will be of interest to the whole genetic researchers community.

Comments/corrections:

1. The authors applied a recently proposed Mendelian randomization (MR) method which requires a large number of mother-child pairs that very few biobanks can provide (at least, at the moment). As a general question regarding this method, is there any information that could be exploited when looking at mothers with more than 1 child? There are some correlations in the PGS between children born from the same mother.

2. As a follow up question to 1 above, was there any adjustment in the association tests being done to account for mothers with more than 1 child (as per Supp Fig 7)? This creates non-independent mother-child pairs.

3. I particularly appreciated the power calculations since this is a novel MR method.

4. The PGS are constructed using GWAS data from the EGG-consortium. Could you add a sentence explaining the transferability, with respect to genetic ancestry, of these PGS to the FinnGen cohort?

5. In Supp Table 1, the PGS labels are not the same as in the Legend.

6. Line 100: It seems that Supp Fig 2 is not related to results discussed.

7. Line 126: Supp Fig 2?

8. In some Figures, change "triangels" for "triangles"

9. Line 342: Supp Tables 7-9

10. Supp Fig 7 is not mentioned (?) in the manuscript.

We thank the reviewers and the editor for the kind and expert comments, resulting in the possibility to resubmit a revised manuscript for further evaluation.

This document contains our replies to the reviewers' comments received and is structured so that **our replies to the reviewer comments are written in blue and the corresponding manuscript changes are written in red color**. We have additionally highlighted all the changes in the accompanying manuscript text file

Below we briefly outline some of the most important changes made to the manuscript based on the feedback from the reviewers. We now:

- a. Present results for two new fetal-specific PGSs for birth weight to support the main conclusion of mainly fetal effects on disease risks (as proposed by the reviewer #1).
- b. Add further triangulation evidence for our findings through the analysis of FinnGen full sibling pairs discordant for the studied disease outcomes (max N = 11,505) (as proposed by the reviewer #2).
- c. Have made an effort to improve the readability of the manuscript. We have changed the nomenclature of the PGSs into more descriptive names (as proposed by the reviewer #1), and the various sensitivity analyses are now presented in their own sections in the Results.

Reviewers' comments:

Reviewer #1 (Remarks to the Author):

Leinonen and colleagues have conducted a Mendelian randomization study to assess the potential causal effects of intrauterine growth on offspring's health outcomes later in life using individual level data from mother-child pairs from FinnGen (n = 36,211). By assessing the effects of genetic scores of birth weight (with either maternal or maternal and fetal effects) using maternal and offspring genotypes, the authors provide supporting evidence that the fetal genome, rather than the intrauterine environment, has long-term effects on health. By using this specific framework, the authors suggest the effect of the fetal polygenic scores of birth weight on adult-life health outcomes reflect genetic pleiotropy rather than causality.

This is a carefully-conducted study that provides valuable information on the lack of causal effects of intrauterine growth on offspring's health outcomes in an expanded set of mother-child pairs compared to previous studies. The evidence provided is solid, although it would benefit from including instruments composed exclusively of genetic variants with fetal only effects on birth weight and assessing its effects on adult health outcomes.

We thank the reviewer for finding our work solid and of interest. The comments received from the reviewer were all excellent and have been very helpful in terms of revising and improving our manuscript.

I have some minor comments:

1. I understand the framing of the study with regards to DOHaD, but I wonder about defining the genetic scores as instruments for “intrauterine growth restriction”, instead of “intrauterine growth”. The publication where these scores were first defined suggest these variants reflect normal variations in birth weight, rather than pathologic cases, and is reflected in the inclusion of individuals with an own birth weight within normal ranges, for instance in the UK Biobank.

We very much appreciate and agree with this comment. As a result, we now refer to intrauterine growth, instead of intrauterine growth restriction, in the context of the genetic scores.

We now omit the term “restriction” in Abstract lines 11 and 12, Introduction lines 38 and 44, Discussion lines 240, 244, 251, 265, 296, 304 and 315, and Methods lines 393 and 413. In addition changes have been made to Figure 1 legend.

2. I also found the naming of the genetic scores slightly confusing, and had to go back and forth between methods/results to check what the acronym referred to. Naming these by the number of variants included, instead of using a descriptive name might be part of the confusion.

This is a very good remark from the reviewer. We had clearly not succeeded in our attempts to name the polygenic scores in the first place. We have now changed the naming of the polygenic scores throughout the manuscript in the following manner that hopefully better highlights which birth weight variants, maternal (M) or fetal (F), are included in each score. We also now include the fetal-specific birth weight polygenic scores in our analyses, as suggested by the reviewer in point 4.

The naming we now use is the following:

Maternal scores:

M-SPECIFIC (previously MBW-29), includes the 29 birth weight variants that act specifically through the maternal genome, unweighted

M-GW (MBW-GW), genome-wide score for maternal effects on birth weight, weighted

Scores with both maternal and fetal variants:

M-ALL (BW-68), includes all birth weight variants (N=68) that have an effect through the maternal genome, some of which have also fetal effects, unweighted

MF-ALL (BW-201), includes all variants (N=201) that have an effect on birth weight either through the maternal or fetal genome or both, unweighted

Fetal scores:

F-SPECIFIC (new), includes the birth weight variants (N=62) that act specifically through the fetal genome, unweighted

F-GW (new), genome-wide score for fetal effects on birth weight, weighted

The updated nomenclature has led to several minor changes throughout the manuscript and supplementary material, which are shown in the manuscript version that highlights the corrections.

3. I appreciate the authors validating the genetic instruments used and assessing the statistical power, which is important. Can the authors comment on how the predictive capacity of the different genetic scores can impact such power analyses?

This is, indeed, a great point to elaborate on. Our power calculations assess the power to capture an effect of a given size measured with respect to a standard deviation change in the polygenic scores for birth weight given the total sample size (mother-child pairs) and proportion of cases for each of the five outcomes. As such these power calculations do not consider the predictive capacity of the scores on birth weight, but are, in simple terms, focused on estimating the power to capture an effect between a normally distributed variable and a binary outcome. The predictive capacity of the scores will, however, be of importance for more detailed interpretation of the effect sizes identified in these analyses.

We now state this limitation in the Results (row 233->):

“In addition, our power calculations do not consider how much variance a PGS explains of birth weight, and hence direct translation of our estimates to clinical effects is more challenging.”

4. While the main conclusion refers largely to the lack of association of the maternal birth-weight increasing alleles on offspring's health outcomes, additional evidence suggesting that fetal genetic scores affect such outcomes would strengthen the conclusion. This could be done by using a genetic score of variants only associated with birth weight (using the weighted effects) in the full set of FinnGen (either individual level data or 2-sample MR), or using the set of variants with only fetal-effects (unweighted effects) in mother-child pairs (combined analysis). This will boost power in analyses where the fetal PGS was not significant after adjusting for the maternal one (lines 150-151).

We thank the reviewer for this suggestion. In the revised version of the manuscript, we include two additional polygenic scores (F-SPECIFIC and F-GW, see answer for point 2 for details) that capture the fetal effects on birth weight. The effects of these polygenic scores are assessed in the full FinnGen data (individual level data) as well as in the mother-child and father-child pairs, and as a new addition also in outcome-discordant sibling pairs from the full FinnGen study sample (see point 1 by reviewer #2).

In short, with the new fetal scores we find:

- As expected, the fetal scores predict own birth weight in full FinnGen and in the parent-child pairs, but predict child birth weight only when the analysis does not include the score in the child as a covariate.
- In the full FinnGen sample, we observe several associations with health outcomes and that these are partly distinct from those seen for the maternal scores. For instance, F-SPECIFIC associates with CHD and statin use similarly to M-SPECIFIC, but displays an association with T2D that is to the opposite direction as for M-SPECIFIC, highlighting the partly distinct biology of the maternal and fetal birth weight variants. Additionally, unlike M-SPECIFIC, F-SPECIFIC does not show a significant association with hypertension, in line with the previously reported observation that maternal component of birth weight shows greater genetic correlation with systolic blood pressure than the fetal one.
- In the mother-child pairs, after adjustment for the maternal genome, F-SPECIFIC shows no significant associations to any of the examined outcomes in the children, while M-SPECIFIC, when in the child, is associated with both CHD and hypertension. The fetal genome-wide score, F-GW, however, shows associations with several outcomes. Notably, the association for T2D is again to the opposite direction as with the maternal scores.
- In the father-child pairs, the associations for the fetal scores are non-significant with the exception of F-GW for T2D in the children. However, the sibling comparisons provide further support for the role of the fetal effects, with significant associations for both CHD and T2D.

As the reviewer suggests, together these results add more weight to the conclusion that genetic pleiotropy in the child genome overall has a more pronounced effect on birth weight - disease associations compared to potential maternal effects.

In response to this comment we have made several changes that are highlighted below:

Added data from two new PGSs (F-SPECIFIC and F-GW) into our manuscript by:

- 1) Updating all the result tables (Please see the Supplementary Dataset)
- 2) Updating our Results section, with a new chapter focused on describing the findings in the mother-child pairs related to the five other birth weight PGSs (rows 166-179)

- 3) Updated the Supplementary Figures to include information and summary statistics of the new PGSs where appropriate (Please see the Supplementary Information document)
- 4) Included description of the construction of the F-SPECIFIC and F-GW PGSs to the Methods (rows 397-419)
- 5) Updated Discussion (row 254->) *“This main conclusion was supported by the numerous sensitivity analyses, including using many different birth weight PGSs...”*

5. In addition to the genetic association, I find it would be informative to describe the observational associations in a similar set of individuals. I acknowledge this is not necessary to answer the main study question.

We agree with the reviewer that this would, indeed, be an informative analysis. We considered this possibility, but, unfortunately, concluded that our data set is not suited for these analyses, as the FinnGen participants for whom birth weight is available are too young for us to see any associations to these health outcomes. Information on birth weight is collected from the archives of the national birth registry (founded in 1987), and is available only for a subset of FinnGen participants born thereafter (Release 10 N=39,578, mean age at the end of follow up = 27.7 yrs.). As the mean age at first diagnosis of the five outcomes studied in full FinnGen is 55.9-59.3 years, it is evident that this birth weight subset lacks power to detect associations to these outcomes.

However, the relationship between birth weight and disease has been previously established in other Finnish datasets, including the Helsinki Birth Cohort Study (HBCS, <https://doi.org/10.1080/07853890.2016.1193786>) and more recently collected cohorts such as the Young Finns Study (YFS, <https://doi.org/10.1097/HJH.0000000000000612>). We now refer to these studies in the text.

To clarify this issue the paragraph starting from line 116 now reads as:

“The connections between birth weight and several diseases are well established at both the epidemiological and genetic levels, also in Finnish datasets^{1,4-7,12,15,26,27}. For the FinnGen participants, measured birth weight data are available only for participants born after 1987. These participants are too young to manifest, e.g., cardiac symptoms, making it unfeasible to study epidemiological patterns in this subset. However, echoing the epidemiological observations, we detected many associations between the birth weight PGSs computed from a person's own genotypes and the disease outcomes in the whole FinnGen dataset (N=412,176).”

6. In line with the reflection in line 319-320, would this be something to consider in the analysis/ discussion of power? From Supp Figure 7, one can see that ~ ¼ of the sample are ~30 years old.

This is a valid point raised by the reviewer. We now state this issue explicitly in the text, both in the results of our power analyses and in the discussion.

In the Results section “Power Calculations” (row 230->)

“These analyses estimate the power given the case frequencies in the available mother-child dataset. We, however, note that although many of the children included into the study are old enough to manifest e.g., cardiac symptoms, a quarter of the children from the mother-child pairs were less than 34.4 yrs old at the end of followup, which is a limiting factor in the dataset.”

In the Discussion (row 312->):

“We also note that the inclusion of children that are below the expected age of onset for some of the included endpoints such as CHD may affect the power to detect effects in our main analyses.”

7. The paragraph starting in line 232 refers mostly to data provided as supplementary material. Would it be worth commenting on these results in the main results’ section?

We understand this comment refers to the result of statin use having a potential postnatal component based on the result from both mother-child and father-child pairs. As a result of this comment and the additional analyses done, we have now separated the results from the father-child pairs into their own section, and extended our discussion around the subject.

In Results (row 196->):

“In line with these expectations, we detected little evidence for paternal effects, except for paternal M-GW associating with statin use (OR 1.06 [95% CI 1.02-1.10], $P=0.00208$), raising the possibility of a postnatal contribution to this phenotype or reflecting the more complex nature of the genome-wide PGSs in terms of tagging genetic pleiotropy (Supplementary Figure 5 and Supplementary Table 6).”

In Discussion (rows 278->):

“Although we could conclude that the effects on disease risks mediated by the birth weight PGSs act principally through the child genome, we detected some instances where the results were less clear-cut. In several instances, we associated the birth weight PGSs with one’s own or child’s risk of receiving statin medication, yet could not always confidently exclude potential maternal or paternal contributions to these associations. This was

especially true in case of the genome-wide PGSs for birth weight (M-GW and F-GW). It thus remains possible that some genetic factors affecting birth weight may be related to statin use through a postnatal, e.g., behavioral, component. Alternatively, the mixed results might partly reflect the complex pleiotropy tagged especially by the genome-wide PGSs. In any case, these results should be followed up and validated in additional datasets.”

8. I find line 262 (conclusion) somewhat confusing: “Our findings, aligned with earlier...”. Later in this sentence, the word “instead” (referring to the previous assertion) and independent effects (referring to maternal transmitted variants) are confusing to me.

The reviewer is correct that the phrasing in the conclusion paragraph was not quite clear. We now have rephrased these sentences and hope the idea on the implications of our findings comes across better in the revised version.

The sentence in the concluding paragraph of the updated Discussion section (row 316->) now reads as:

“In contrast to the role of intrauterine conditions, our findings support a model wherein the genetic factors within the maternal genome that influence the child's birth weight, contribute to the child's disease risks only when transferred to the child's genome.”

9. The construction of the polygenic score of gestational duration is not explained in methods

We thank the reviewer for spotting this and apologize for the mistake. The construction of the polygenic score for gestational duration is now included in the methods section:

Methods (rows 451-454):

“The polygenic scores for gestational length were constructed based on GWAS results from the EGG-consortium, using summary statistics from maternal GWAS meta-analysis of gestational duration based on 151,987 women¹⁷. The genome-wide PGS for gestational length used in the analyses was calculated similarly as the genome-wide scores for birth weight, using PRS-CS and the FinnGen PRS pipeline.”

10. The genetic scores have a different naming in Supp Table 1/2 than in the main text.

We thank the reviewer for this observation, which reflects our struggle to find appropriate names for the PGS for the previous version of the manuscript. We have now fixed this, ensuring that we use the new, hopefully more descriptive names for the PGSs consistently throughout the manuscript.

Several minor changes throughout the manuscript and supplementary material, please see the manuscript version with author corrections highlighted.

11. I appreciate very much that the authors added the code for running the simulations. I have rerun it, and noticed that there is a missing closing bracket in the 3rd row by the end of the code. Otherwise, it works perfectly well and power analyses are easy to reproduce.

We are delighted to hear the reviewer found the code helpful and thank them for spotting the typo in the script.

We have fixed the said missing bracket in the code.

Reviewer #2 (Remarks to the Author):

This study found limited evidence of a causal relationship between intrauterine growth restriction (IUGR) and non-communicable diseases such as coronary heart disease (CHD) and hypertension in the FinnGen biobank. Instead, the findings suggest that genetic pleiotropy in the child is mostly responsible for the association of birth weight and the five investigated diseases. Using 4 different polygenic scores (PGS) and sensitivity analyses, the conclusions and interpretation are solid and highly convincing. The abstract, introduction and discussion are well written, and previous literature clearly exposes the current knowledge about the developmental origins of health and disease (DOHaD hypothesis) conundrum. These results will be of interest to the whole genetic researchers community.

We thank the reviewer for this positive overview of our work. The insightful comments received from the reviewer, particularly those regarding the sibling data, have been truly helpful in terms of revising and improving our manuscript.

Comments/corrections:

1. The authors applied a recently proposed Mendelian randomization (MR) method which requires a large number of mother-child pairs that very few biobanks can provide (at least, at the moment). As a general question regarding this method, is there any information that could be exploited when looking at mothers with more than 1 child? There are some correlations in the PGS between children born from the same mother.

We thank the reviewer for the excellent suggestion to take further advantage of the distinct characteristics of the available data set. The 36,211 mother-child pairs are, indeed, not fully independent as 5,483 mothers have more than one genotyped child included in the analyses. These non-independent pairs can, indeed, allow for additional within-sibling

investigations that can provide further triangulation evidence for the (lack of) maternal associations, and potential fetal contribution to disease as described in this manuscript.

In our case, all siblings are concordant for the maternal exposure (M-SPECIFIC), but can be discordant for the outcome and additionally differ in terms of the number of transmitted alleles of M-SPECIFIC that increase birth weight when in the maternal genome. This setting can allow, for instance, the examination of the association of M-SPECIFIC in the children, which in the primary analyses we found associated with CHD and hypertension, with the outcomes of interest under a matched case-control setting.

The data available for such analyses within the FinnGen mother-child pairs is, unfortunately, a bit limited in size with, e.g., only 362 full sibling pairs that are discordant for CHD. With this data set, using conditional logistic regression, we reassuringly recapitulate a similar sized effect of M-SPECIFIC in the children with CHD and hypertension (for CHD -0.117 (0.147), $p=0.42$, hypertension ($N=1729$) -0.063 (0.068) $p=0.36$) as in the main analysis, but the associations remain non-significant ($p>0.05$). As such, these results are more speculative and we do not consider these add much value to the current manuscript.

We can, however, extend these analyses to the full FinnGen data set, where we have a much larger number of sibling pairs available. By selecting sex-matched full sibling pairs from FinnGen R10, we identified between 6,016 (CHD) and 11,505 (statin use) pairs discordant for the five outcomes of interest available for these analyses. The age range of this set of samples is different to the children of the mother-child pairs, with an average age of 63.9, which is subsequently reflected in the higher case frequencies in this sibling subset.

When analyzed in the sibling set in the full FinnGen data using a conditional logistic regression model, M-SPECIFIC associates ($p<0.05$) with CHD ($N=6,016$) and hypertension ($N=6,164$) with directionally consistent effect estimates to the main analyses (CHD: effect = -0.072 , se = 0.026 , p-value = 0.00487 ; hypertension: effect = -0.065 , se = 0.027 , p-value = 0.0172). These results, therefore, provide further validation for the finding that the maternal birth weight variants have pleiotropic effects in the child genome that contribute to the observed links between birth weight and later life disease risk.

As a result of this suggestion, we have now added data on the PGSs effects in siblings into our manuscript. To incorporate these sibling findings into the introduction, results, methodology, and discussion of the manuscript, we have made several changes to the respective sections.

The findings are now summarized in Results under section “Analysis of FinnGen full sibling pairs strengthens the conclusions from parent-child data”, as follows (row 204->) :

“We finally sought validation of these findings using the 63,428 full sibling pairs (mean age 63.9 yrs. [SD 16.4]) from FinnGen. Leveraging sibling pairs discordant for the outcomes of interest enables another nuanced way to determine whether a PGS for maternal effects on birth weight affects disease risk through the maternal or fetal genome. As siblings share their maternal genetic effects on birth weight, any association of the maternal PGS with disease risk seen in these sibling analyses should solely reflect the child’s own genetic effects. To test this, we assessed associations of the birth weight PGSs in a matched case-control setting using same-sex sibling pairs, discordant for disease status (case numbers ranging from 6,016 CHD cases to 11,505 statin users in the final datasets). Applying a conditional logistic regression model, we found that one’s own PGSs for maternal effects on birth weight is associated with a reduced risk of CHD and hypertension in the full sibling pairs (M-SPECIFIC OR=0.93 [95%CI 0.89-0.98], P=0.00487 and OR=0.94 [95% CI 0.89-0.99], P=0.0172, for CHD and hypertension, respectively) (Supplementary Table 7 and Supplementary Figure 6). Similarly, fetal-specific birth weight variants displayed an effect on CHD through one’s own genome (OR=0.95 [95% CI 0.89-0.99], P=0.0105 for F-SPECIFIC). In contrast, besides the fetal genome-wide PGS (F-GW) association to statins (OR=0.92 [95% CI 0.89-0.96], P=5.0e-05, none of the PGSs containing variants with maternal effects on birth weight were associated with statin use or cancer risk in siblings. However, we observed distinct connections for the birth weight PGSs to T2D, with the direction of the effect depending on whether the PGS tags maternal or fetal effects (OR=1.06 [95% CI 1.01-1.11], P=0.0234 for M-GW and OR=0.89 [95% CI 0.85-0.93], P=1.6e-06 for F-GW) (Supplementary Table 7 and Supplementary Figure 6). Overall, these sibling comparisons further support the interpretation that the effects of birth weight PGSs on disease are primarily mediated by genetic pleiotropy of the birth weight variants in the child’s genome”

The sibling results are found in full detail in Supplementary Table 7, and are presented in Supplementary Figure 6. For description of the other changes made based on this comment, please refer to Introduction rows 60 and 70, Discussion rows 254->, 271->, 291->, and Methods rows 370->, 381->, and 456.

2. As a follow up question to 1 above, was there any adjustment in the association tests being done to account for mothers with more than 1 child (as per Supp Fig 7)? This creates non-independent mother-child pairs.

This is another excellent point raised by the reviewer. We acknowledge the potential bias that such non-independency in the mother-child pairs generates, and therefore had conducted a sensitivity analysis where only one child per mother was included (N=28,792) using the same framework (results shown in Supplementary Table 5 and mentioned in the results section under “Sensitivity analyses in mother-child pairs support lack of maternal effects on child disease risk”). We appreciate this validation step is an important aspect and in the revised version of the manuscript elaborate these results and their implications as

indicated below in the manuscript changes. Reassuringly the results and their interpretation stay similar between the primary (all children included) and the sensitivity (only one child included) analysis. In particular, the association for CHD through the child genome for M-SPECIFIC (i.e., the PGS previously called MBW-29) remains unchanged (associations for the PGS in the child after adjustment for the PGS in the mother, all children: effect = -0.135, se = 0.039, p-value = 0.000470; one child: effect = -0.146, se = 0.026, p-value = 0.00154). The above sensitivity analysis approach of leaving all but one child out is admittedly a simplistic one, yet given the size of the data set does not result in considerable losses in power.

We have now clarified the way we present the several different sensitivity analyses in the main manuscript by dividing these analyses into different subsections in the Results.

We now touch upon this issue of non-independent mother-child pairs in the opening paragraph of the Results section “Sensitivity analyses in mother-child pairs support lack of maternal effects on child disease risk” (row 180->):

“As the FinnGen mother-child data contains mothers (N=5,483) that have more than one child included in the dataset, we tested how excluding such non-independent pairs affects the detection of the maternal effects on disease.”

Moreover, we have clarified the way we describe these analyses, including only one child per mother, in the Methods (row 447->) :

“As further sensitivity analyses, we a) performed similar analyses using only one (oldest) child per mother (N=28,582), b) kept only the oldest mother and her oldest child from an extended family (N=22,454 mother-child pairs with 3th degree relatives and closer for both mothers and children removed from the analysis)”

3. I particularly appreciated the power calculations since this is a novel MR method.

We are glad to hear the reviewer found the power calculations helpful.

4. The PGS are constructed using GWAS data from the EGG-consortium. Could you add a sentence explaining the transferability, with respect to genetic ancestry, of these PGS to the FinnGen cohort?

We appreciate that commenting on the transferability of the PGSs to FinnGen is important. Finns are a genetically unique population largely due to the recent bottlenecks. This has a profound effect on the frequency of genetic variants that are rare (<https://doi.org/10.1371/journal.pgen.1004494>), whereas the allele frequencies of common variants are largely similar when comparing Finland to other European countries

(<https://doi.org/10.1038/ejhg.2016.205>). We observed that the concordance of the birth weight associated allele frequencies between the discovery GWAS (EGG Consortium) vs. our study is really high ($R=0.981$), reflecting the latter observation. The authors of the discovery GWAS also showed that the variants identified in the GWAS explain a substantial proportion of variance in birth weight in another North European population (MoBa from Norway). Hence, we generally expected the transferability of the PGSs to FinnGen to be relatively high. This proved to be the case as shown by all the PGSs predicting birth weight in the FinnGen in an expected manner. We now comment on the transferability in the methods.

Methods (row 429->):

“The PGSs used in this study are derived from a trans-ancestral GWAS meta-analysis, primarily composed of individuals of European ancestry, with the variants identified in the GWAS explaining a substantial proportion of variance in birth weight in the Norwegian MoBa cohort¹⁵. In line with these findings from another Northern European cohort, we observed that the constructed unweighted and genome-wide PGSs showed transferability also to the FinnGen cohort, capturing the desired effects on birth weight (Supplementary Table1).”

5. In Supp Table 1, the PGS labels are not the same as in the Legend.
6. Line 100: It seems that Supp Fig 2 is not related to results discussed.
7. Line 126: Supp Fig 2?
8. In some Figures, change “triangels” for “triangles”
9. Line 342: Supp Tables 7-9
10. Supp Fig 7 is not mentioned (?) in the manuscript.

We thank the reviewer for spotting these mistakes. We have gone through all the suggested edits, and the observed inconsistencies and typos have been fixed in the new version of the manuscript.

REVIEWERS' COMMENTS:

Reviewer #1 (Remarks to the Author):

I'm happy with the responses to my points and have no further comments. Congratulations on a great manuscript.

Reviewer #2 (Remarks to the Author):

I would to thank the authors for their additional analyses which bring more evidence and strengthen their initial claim. The revised manuscript has improved clarity, especially with regard to the names of the different PGS, over the previous version.